# CNN2 silencing inhibits colorectal cancer development through promoting ubiquitination of EGR1

Jinghu He[1],* , Xiaohong Yang[1],*, Chuansen Zhang[2],*, Ang Li[1], Wei Wang[1], Junjie Xing[1], Jifu E[1], Xiaodong Xu[1], Hao Wang[1], Enda Yu[1], Debing Shi[3,4], Hantao Wang[1]

Colorectal cancer (CRC) is one of the most commonly diagnosed malignant tumors of the digestive tract. H2-calponin (CNN2), an actin cytoskeleton-binding protein, is an isoform of the calponin protein family whose role in CRC is still unknown. Research based on clinical samples showed the up-regulation of CNN2 in CRC and its association with tumor development, metastasis, and poor prognosis of patients. Both in vitro loss-of-function and gain-of-function experiments showed that CNN2 participates in CRC development through influencing malignant cell phenotypes. In vivo, xenografts formed by CNN2 knockdown cells also showed a slower growth rate and smaller final tumors. Furthermore, EGR1 was identified as a downstream of CNN2, forming a complex with CNN2 and YAP1 and playing an essential role in the CNN2-induced regulation of CRC development. Mechanistically, CNN2 knockdown down-regulated EGR1 expression through enhancing its ubiquitination, thus decreasing its protein stability in a YAP1-dependent manner. In summary, CNN2 plays an EGR1-dependent promotion role in the development and progression of CRC, which may be a promising therapeutic target for CRC treatment.

## Introduction

Colorectal cancer (CRC) is one of the most commonly diagnosed malignant tumors of the digestive tract, characterized by high morbidity and mortality, and currently ranks third in incidence and second in mortality among all malignant tumors (Bray et al, 2018; Siegel et al, 2021). In China, the incidence and mortality of CRC rank fifth and are still increasing year by year (Chen et al, 2016; Cao et al, 2020). Surgical resection is the most classical and effective modality for the treatment of CRC. However, because of the insidious onset, rapid progression, and susceptibility to metastasis of CRC, surgical resection cannot achieve complete removal of the tumor in most cases (Keum & Giovannucci, 2019). In recent years, although more

and more new treatment modalities such as targeted therapy and immunotherapy have begun to be applied in clinical practice, which is benefited from the progress of the CRC diagnosis and treatment technology, the prognosis and life quality of CRC patients are still far from reaching a satisfactory degree (Ganesh et al, 2019; Katona & Weiss, 2020). At the molecular level, the development of CRC is associated with dysregulation or dysfunctions of many related genes. Therefore, an in-depth understanding of the molecular mechanisms of the pathogenesis, progression, and metastasis of CRC can help to develop effective targeted treatment strategies and improve the quality of life and prognosis of patients (Lech et al, 2016).

H2-calponin (CNN2), an actin cytoskeleton-binding protein, is an isoform of the calponin protein family, which also includes basic H1-calponin (CNN1) and acidic H3-calponin (CNN3) (Liu & Jin, 2016). H1-calponin is specifically expressed in smooth muscle cells, whereas CNN2 is expressed in smooth muscle cells and in a variety of other cells (Wu & Jin, 2008; Plazyo et al, 2019). The functional study of CNN2 carried out by Moazzem Hossain et al revealed that infecting smooth muscle cells with the cDNA sense or antisense strand of CNN2 by stable transfection or cis-transfection was able to inhibit the proliferation and division of the cells. Further studies clarified that CNN2 was mainly recruited on the nuclear ring formed by actin microfilaments in dividing binucleated cells, indicating that the regulatory effects of CNN2 on smooth muscle cell proliferation were carried out by regulating actin skeleton activity (Hossain et al, 2003). In addition to muscle cells, CNN2 also shows some regulatory role in tumor cell phenotypes. However, existing studies are fairly limited and only preliminarily reveal its regulatory role in several types of malignant tumors such as gastric cancer, liver cancer, and prostate cancer (Moazzem Hossain et al, 2014; Hu et al, 2017; Kang et al, 2018). Its association with CRC remains to be developed.

Herein, the role played by CNN2 in the development and progression of CRC was studied at the clinical level, cellular level, and animal level. The detection or statistical analysis of CNN2 expression was performed in either the local patient cohort or The

---

[1]Department of General Surgery, Changhai Hospital Affiliated to Navy Medical University, Shanghai, China  [2]Department of Anatomy, Naval Medical University, Shanghai, China  [3]Department of Colorectal Surgery, Fudan University Shanghai Cancer Center, Shanghai, China  [4]Department of Oncology, Shanghai Medical College, Fudan University, Shanghai, China

Correspondence: hantaowang@126.com; shidebing7819@163.com
*Jinghu He, Xiaohong Yang, and Chuansen Zhang contributed equally to this work

Cancer Genome Atlas (TCGA) database. In vitro loss-of-function and gain-of-function experiments were conducted for displaying the regulatory functions of CNN2 in CRC cell phenotypes, followed by in vivo investigation and verification. All the results indicated the promotion effects of CNN2 on CRC development and the inhibitory effects of CNN2 knockdown. Moreover, the mechanism exploration revealed EGR1 as a potential downstream target for CNN2 to mediate the CRC regulation.

# Results

### CNN2 is up-regulated in CRC and associated with disease development

A tissue microarray, containing 90 CRC tumor tissues and 78 para-carcinoma normal tissues, was used for IHC analysis to show the expression pattern of CNN2 in CRC. As shown in Fig 1A and B and Table 1, the expression of CNN2 in tumor tissues was significantly higher than in normal tissues. Further comparison of the representative images (Fig 1A) and statistical analysis (Tables 2 and S1) suggested that the up-regulated CNN2 in CRC may be associated with tumor development and lymphatic metastasis. Actually, the up-regulation of the CNN2 mRNA level in CRC could also be observed in data collected from TCGA (Fig 1C). Specifically, CNN2 was also the only one with up-regulation in CRC tissues among the three members of the CNN family (Figs 1C and S1). Finally, a Kaplan–Meier survival analysis indicated CNN2 as a potential biomarker for the worse prognosis of CRC patients (Fig 1D).

### Silencing CNN2 inhibits CRC development in vitro

For verifying the proposed role of CNN2 in CRC, shRNAs were prepared for silencing CNN2, of which the more efficient ones (shCNN2-2 and shCNN2-3) were selected for constructing HCT116 and RKO cell models with CNN2 knockdown (Fig S2). After checking the transfection efficiency by fluorescence imaging (Fig S3) and verifying the CNN2 knockdown by qRT-PCR and Western blotting (Fig 2A), detection of phenotypes of HCT116 and RKO cells transfected with shCtrl or shCNN2 was performed. As shown in Fig 2B, the observation of cell growth clearly manifested the inhibited cell proliferation of CRC cells after the knockdown of endogenous CNN2 expression. Not surprisingly, results of flow cytometry exhibited the significantly enhanced cell apoptosis (Fig 2C). Moreover, considering the close association between CNN2 expression and lymphatic metastasis of CRC, cell migration ability was evaluated by both the wound-healing assay and the Transwell assay, showing the suppressed cell motility upon CNN2 knockdown (Fig 2D and E). Collectively, all these results showed the regulatory effects of CNN2 knockdown on CRC phenotypes, which agrees with our previous outcomes.

### Silencing CNN2 inhibits CRC development in vivo

The inhibitory role of CNN2 knockdown in CRC development was subsequently validated using mouse xenograft models constructed by subcutaneous injection of RKO cells in both shCtrl and shCNN2 groups. As shown in Fig 3A, xenografts in the shCtrl group were always larger than those in the shCNN2 group during the whole tumor growth period. According to the in vivo fluorescence imaging made before euthanizing mice, xenografts in the shCtrl group truly showed more vigorous growth activity (Fig 3B). Upon euthanizing the model mice and collecting the xenografts, direct evidence could be obtained from the macroscopic observation of tumor size and tumor weight (Fig 3C); indirect evidence was shown by the IHC analysis of Ki-67 expression, which is a representation of tumor growth, in sections of xenografts (Fig 3D).

### CNN2 may regulate CRC development through affecting EGR1 expression

For further getting insight into the mechanism by which CNN2 regulates the development of CRC, gene expression profiling of shCtrl and shCNN2 RKO cells was acquired and analyzed. Differentially expressed genes were identified based on the fold change of the mean of expression (fold change ≥ 1.3) and FDR (<0.05) from the P-value calculated based on the linear model of the empirical Bayesian distribution. In total, 523 up-regulated genes and 648 down-regulated genes were characterized (Figs 4A and S4A), based on which were performed enrichment analyses of the canonical signaling pathway and the disease/function using the Ingenuity Pathway Analysis (Fig S4B and C). On this basis, a CNN2-centered molecular interaction network was built (Fig S5A) and several promising downstream targets for CNN2 were selected for further verification by qRT-PCR and Western blotting (Fig S5B–D). Among the several candidates with down-regulated mRNA and protein expression in CNN2 knockdown cells, EGR1 was found to be the only one with co-expression profile with CNN2 based on the analysis of TCGA database (Fig S5E). Accordingly, EGR1 was also found to be up-regulated in CRC tissues compared with normal tissues and thus considered as a potential downstream of CNN2 (Fig 4B). Moreover, the down-regulation of the EGR1 protein level in HCT116 and RKO cells with CNN2 knockdown was also further verified and is shown in Fig S6.

### CNN2 may regulate ubiquitination and the expression of EGR1 in a YAP1-dependent manner

For uncovering the regulatory mechanism, a series of co-immunoprecipitation assays were performed for detecting the protein–protein interaction. The existence of CNN2 and EGR1 in complexes precipitated by anti-Flag (CNN2-Flag) or anti-EGR1 suggested the interaction between CNN2 and EGR1 (Fig 4C and D). Similarly, interactions between CNN2 and YAP1 and between YAP1 and EGR1 were also visualized by co-immunoprecipitation (co-IP) assays (Fig 4C and E). Therefore, it was supposed that CNN2 may regulate EGR1 expression through forming the CNN2/YAP1/EGR1 complex. Subsequently, for exploring the detailed route by which CNN2 regulates EGR1 expression, we investigated the effects of CNN2 knockdown on protein stability of EGR1. The results showed that upon the treatment of cycloheximide to prohibit protein synthesis, the knockdown of CNN2 distinctly decreased EGR1 protein stability and down-regulated its expression (Fig 5A). Interestingly, the

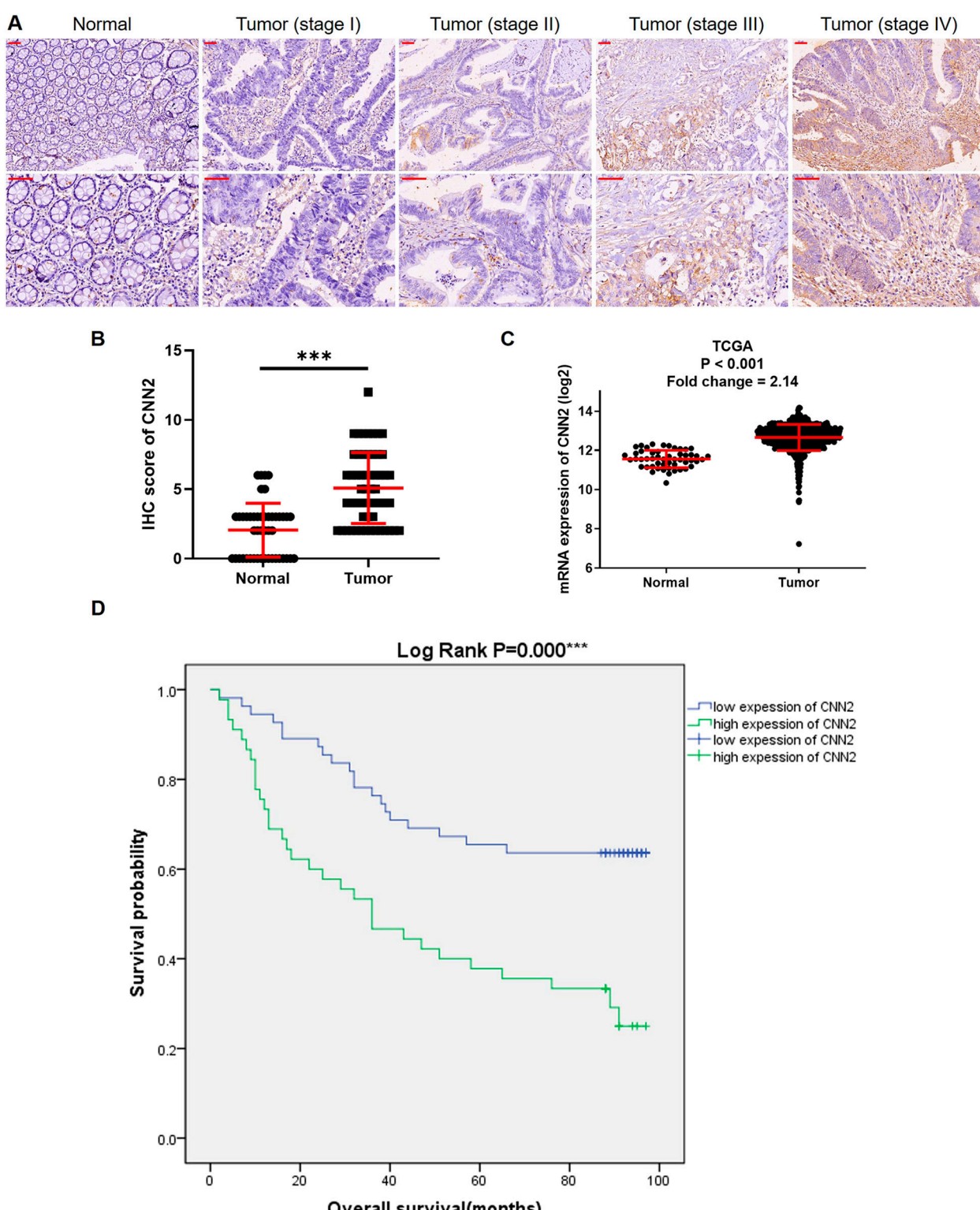

**Figure 1. Expression of CNN2 is up-regulated in CRC.**
**(A)** Representative images of IHC analysis of CNN2 expression in normal tissues and CRC tissues with different pathological stages. Scale bar = 50 $\mu$m. **(B)** IHC scores of CNN2 in all normal and CRC tissues were summarized and statistically analyzed. **(C)** Expression of CNN2 in normal and CRC tissues collected from TCGA database was collected and analyzed. **(D)** Kaplan–Meier analysis was used for revealing the association between CNN2 expression and CRC patients' prognosis. ***$P < 0.001$. Source data are available for this figure.

**Table 1.   Expression patterns of CNN2 in colorectal tissues and normal tissues revealed in immunohistochemistry analysis.**

| CNN2 expression | Tumor tissue | | Normal tissue | |
|---|---|---|---|---|
| | Cases | Percentage | Cases | Percentage |
| Low | 55 | 55.0% | 67 | 85.9% |
| High | 45 | 45.0% | 11 | 14.1% |

*P* < 0.001.

**Table 2.   Relationship between CNN2 expression and tumor characteristics in patients with colorectal cancer.**

| Features | No. of patients | CNN2 expression | | *P*-Value |
|---|---|---|---|---|
| | | Low | High | |
| All patients | 100 | 55 | 45 | |
| Age (yr) | | | | 0.238 |
| ≤68 | 51 | 31 | 20 | |
| >68 | 49 | 24 | 25 | |
| Gender | | | | 0.435 |
| Male | 49 | 25 | 24 | |
| Female | 51 | 30 | 21 | |
| Tumor size | | | | 0.928 |
| <5 cm | 41 | 23 | 18 | |
| ≥5 cm | 58 | 32 | 26 | |
| Grade | | | | 0.762 |
| I | 1 | 0 | 1 | |
| II | 83 | 46 | 37 | |
| III | 15 | 9 | 6 | |
| IV | 1 | 0 | 1 | |
| Stage | | | | 0.003 |
| 1 | 6 | 3 | 3 | |
| 2 | 54 | 38 | 16 | |
| 3 | 39 | 14 | 25 | |
| 4 | 1 | 0 | 1 | |
| T-infiltrate | | | | 0.742 |
| T1 | 1 | 0 | 1 | |
| T2 | 5 | 3 | 2 | |
| T3 | 74 | 41 | 33 | |
| T4 | 19 | 11 | 8 | |
| Lymphatic metastasis (N) | | | | 0.001 |
| N0 | 60 | 41 | 19 | |
| N1 | 30 | 11 | 19 | |
| N2 | 10 | 3 | 7 | |

regulation of EGR1 by CNN2 knockdown could be almost completely eliminated after treating cells with MG132 (Fig 5B), a proteasome inhibitor, indicating that the influence on EGR1 protein stability by CNN2 knockdown involves the ubiquitin–proteasome system. Indeed, the

examination of EGR1 ubiquitination showed that CNN2 knockdown apparently increased the ubiquitination modification of EGR1 (Fig 5C), which, in combination with the above results, proved that CNN2 may regulate EGR1 expression through the ubiquitin–proteasome system. More importantly, to clarify the significance of the formation of the CNN2/YAP1/EGR1 complex in the CNN2-induced regulation of EGR1, further protein stability tests were performed. As shown in Fig 5D and E, YAP1 not only can regulate the stability of the EGR1 protein, but also plays a decisive role in the regulation of EGR1 stability caused by CNN2. Collectively, we deduced that CNN2 may regulate ubiquitination and the expression of EGR1 in a YAP1-dependent manner.

### The regulatory effects of CNN2 on CRC are EGR1-dependent

Given EGR1 as a potential downstream of CNN2, further "rescue" experiments were performed to show the synergistic effects of CNN2 and EGR1 on cell phenotypes of CRC. For this purpose, a CNN2 overexpression construct (the CNN2 group and Vector as the negative control group) was generated and used for RKO cell transfection together with shEGR1 (targeting EGR1 silence; Fig S7) (Fig S8). As shown in Fig 6A, enhanced proliferation of CRC cells induced by CNN2 overexpression and suppressed cell growth by EGR1 knockdown could be clearly observed. On the contrary, the promotion of cell proliferation by CNN2 overexpression could be reversed by the simultaneous knockdown of EGR1, indicating the key mediator role played by EGR1. Similar results could also be observed in the colony formation assay (Fig 6B). Conversely, although relatively slightly, CNN2 overexpression suppressed cell apoptosis, which could also be partially recovered by EGR1 knockdown, whereas EGR1 knockdown significantly increased cell apoptosis (Fig 6C). Finally, the synergistic effects of CNN2 and EGR1 on cell migration were evaluated by the Transwell assay, displaying CNN2 overexpression–induced motility enhancement, EGR1 knockdown–induced motility decline, and the synergistic inhibition of cell migration induced by simultaneous CNN2 over-expression and EGR1 knockdown (Fig 6D). Similarly, HCT116 and RKO cell models with CNN2 knockdown and EGR1 overexpression were also constructed (Figs S9 and S10). It was demonstrated that the inhibitory effects on cell proliferation and migration and the promotion effects on cell apoptosis induced by CNN2 knockdown could be partially reversed by EGR1 overexpression (Figs S11–S13). All the above results indicated the key role of EGR1 as a mediator in the CNN2-induced regulation of CRC. Notably, a schematic figure was drawn to display the mechanism by which CNN2 regulates CRC development.

## Discussion

In this study, a novel regulator in the development and progression of CRC, known as CNN2, was identified on multiple levels. As a member of the calponin family, the role of CNN2 as a regulator in actin cytoskeleton functions has been well investigated (Liu & Jin, 2016). More importantly, unlike CNN1 that is only expressed in muscle cells, CNN2 exists and possesses specific functions in a variety of cells. For example, Jin et al demonstrated that the down-

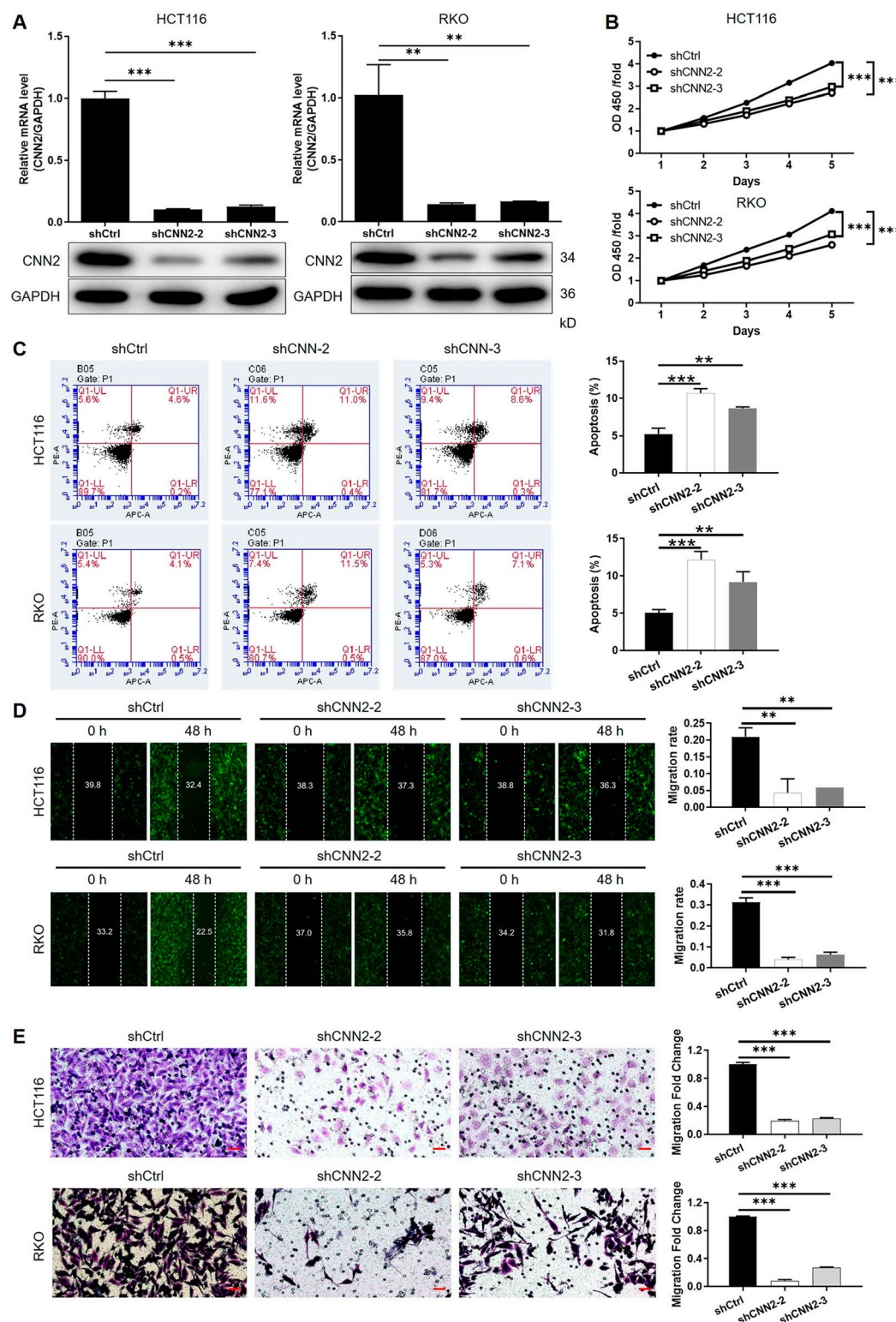

**Figure 2. CNN2 knockdown inhibits CRC development in vitro.**
**(A)** qRT-PCR and Western blot were performed for evaluating the knockdown efficiencies of CNN2 in HCT116 and RKO cells. **(B)** Cell proliferation of HCT116 and RKO cells in shCtrl and shCNN2 groups was detected by the CCK8 assay. **(C)** Apoptotic cell percentage of HCT116 and RKO cells in shCtrl and shCNN2 groups was detected by flow cytometry. **(D, E)** Cell migration of HCT116 and RKO cells in shCtrl and shCNN2 groups was assessed by the wound-healing assay (D) and the Transwell assay (E), respectively. Scale bar = 50 μm. Data were shown as the mean ± SD. *$P < 0.05$, **$P < 0.01$, and ***$P < 0.001$.
Source data are available for this figure.

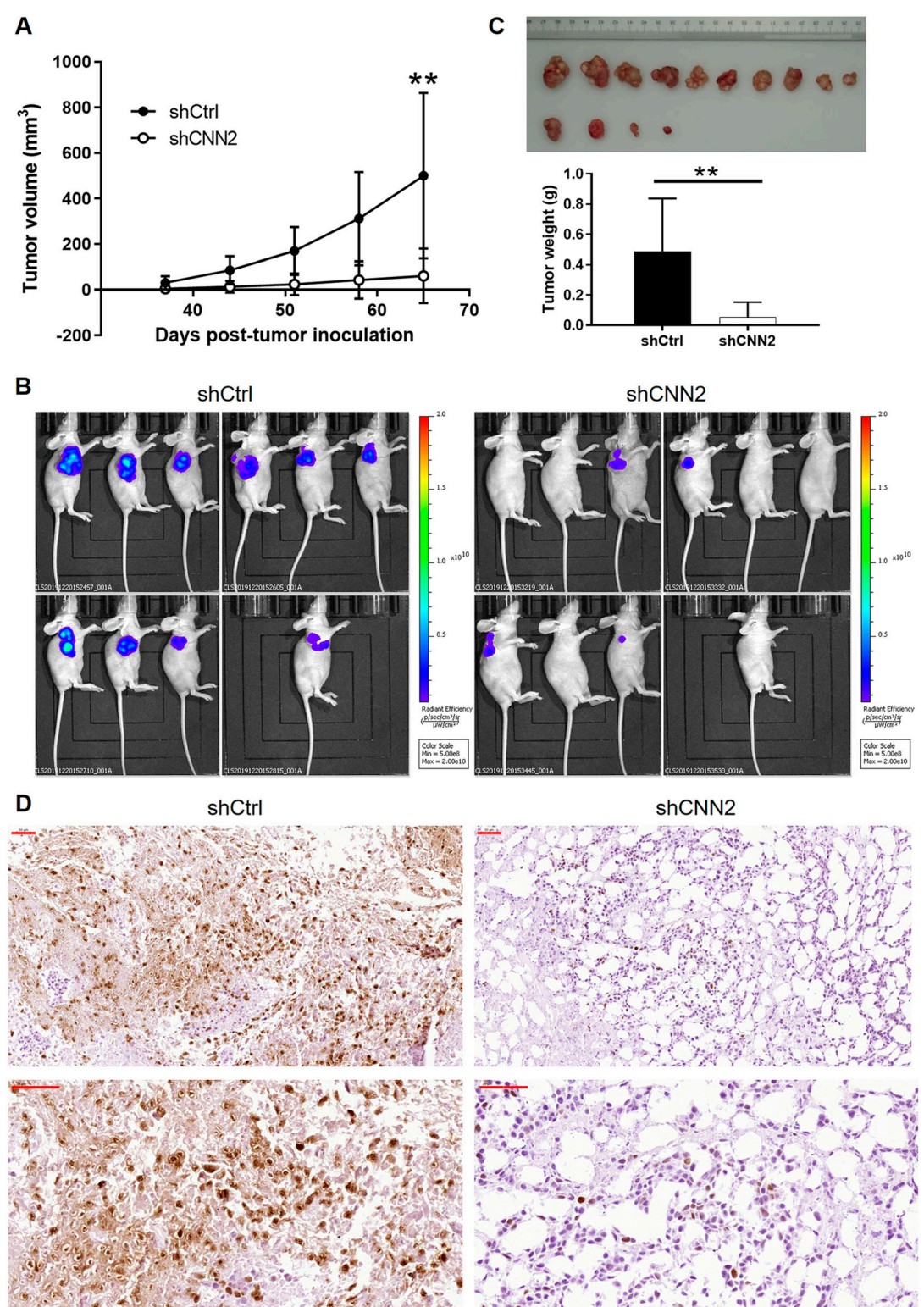

**Figure 3. CNN2 knockdown inhibits CRC development in vivo.**
Mouse xenograft models were constructed through the injection of RKO cells transfected with shCtrl or shCNN2. **(A)** Differential tumor volume between shCtrl and shCNN2 groups indicated the inhibited tumor growth in the shCNN2 group. **(B)** Before animal euthanizing, in vivo fluorescence imaging was performed to monitor the growth and metastasis of xenografts. **(C)** After euthanizing mice, xenografts were removed and collected for taking photographs and weighing. **(D)** Expression of Ki-67 in xenografts of shCtrl and shCNN2 groups was detected by IHC analysis. Scale bar = 50 $\mu$m. Data were shown as the mean ± SD. **$P$ < 0.01.
Source data are available for this figure.

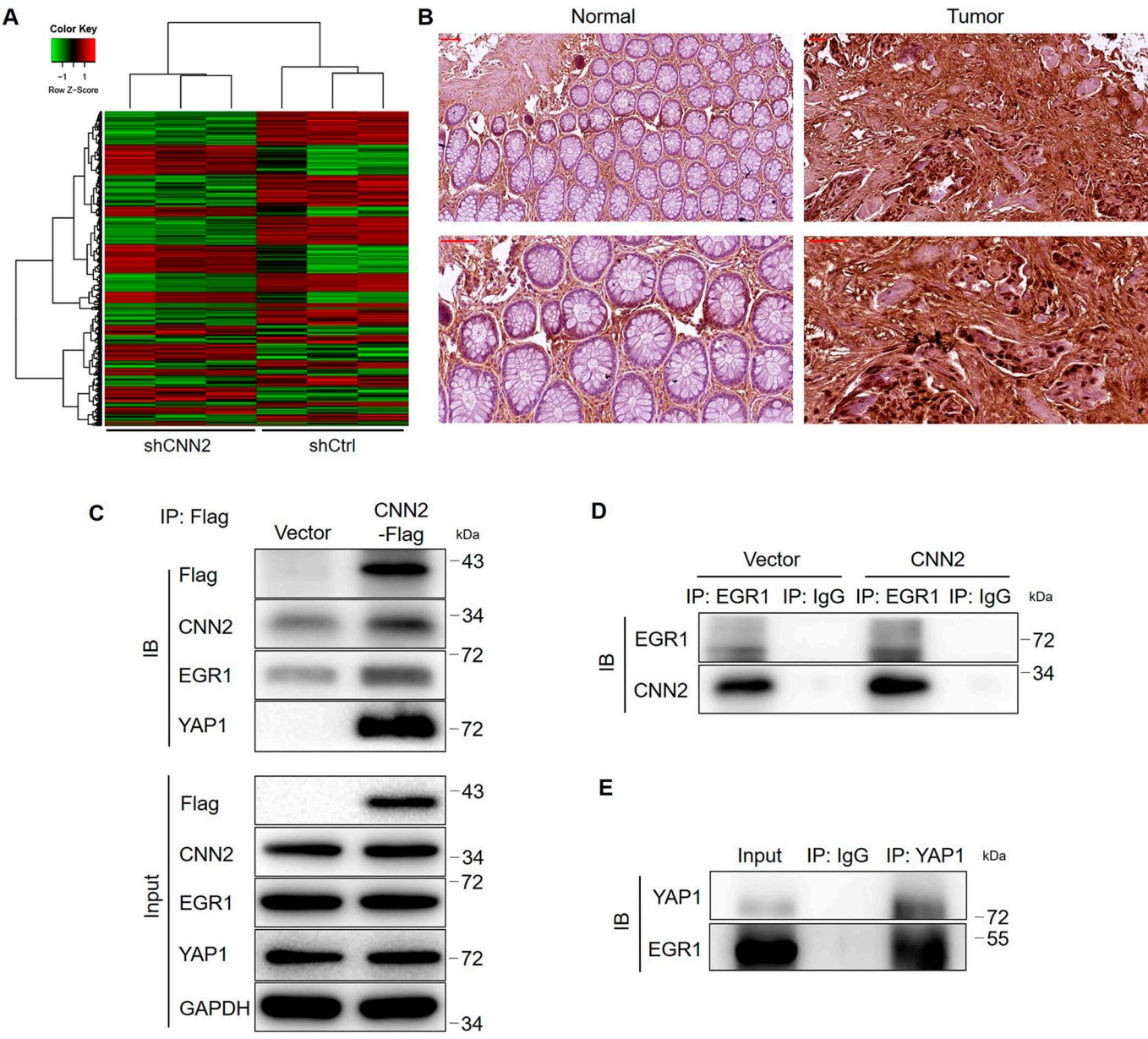

**Figure 4.  Exploration of the downstream mechanism of CNN2 in CRC.**
**(A)** Gene microarray was performed to obtain the gene expression profiling of shCtrl and shCNN2 RKO cells, and the heat map was shown. **(B)** Expression of EGR1 in normal and CRC tissues was detected by IHC analysis. **(C, D, E)** Total proteins obtained from RKO cells overexpressing Flag-tagged full-length CNN2 were subjected to immunoprecipitation with an anti-Flag antibody (C), anti-EGR1 antibody (D), or anti-YAP1 antibody (E), followed by Western blot with indicated antibodies. Scale bar = 50 µm. Data were shown as the mean ± SD. ***P < 0.001.
Source data are available for this figure.

regulation of CNN2 in macrophages may be considered as a new cytoskeleton-based mechanism that may alter cytokine processing and secretion through ER stress, modulate immune responses, and promote quiescence for the treatment of inflammatory diseases such as inflammatory arthritis (Huang et al, 2016; Plazyo et al, 2019). CNN2 was also reported to be capable of regulating functions of myofibroblasts, thus highlighting its potential in the treatment and prevention of the calcific aortic valve disease (Plazyo et al, 2018). Besides, recently, accumulating evidence showed that CNN2 may

take part in the development of malignant tumors because of its dysregulation and dysfunctions. In pancreatic ductal adenocarcinoma, the high expression of CNN2 was found to be significantly associated with less lymph node metastasis and longer survival of patients, suggesting its tumor-suppressing functions (Qiu et al, 2017). Conversely, a lot more studies revealed the role of CNN2 in human cancers as a tumor-promoting molecule. Zhou et al detected the higher expression of CNN2 in the plasma and tissues of patients with breast cancer compared with healthy ones, indicating

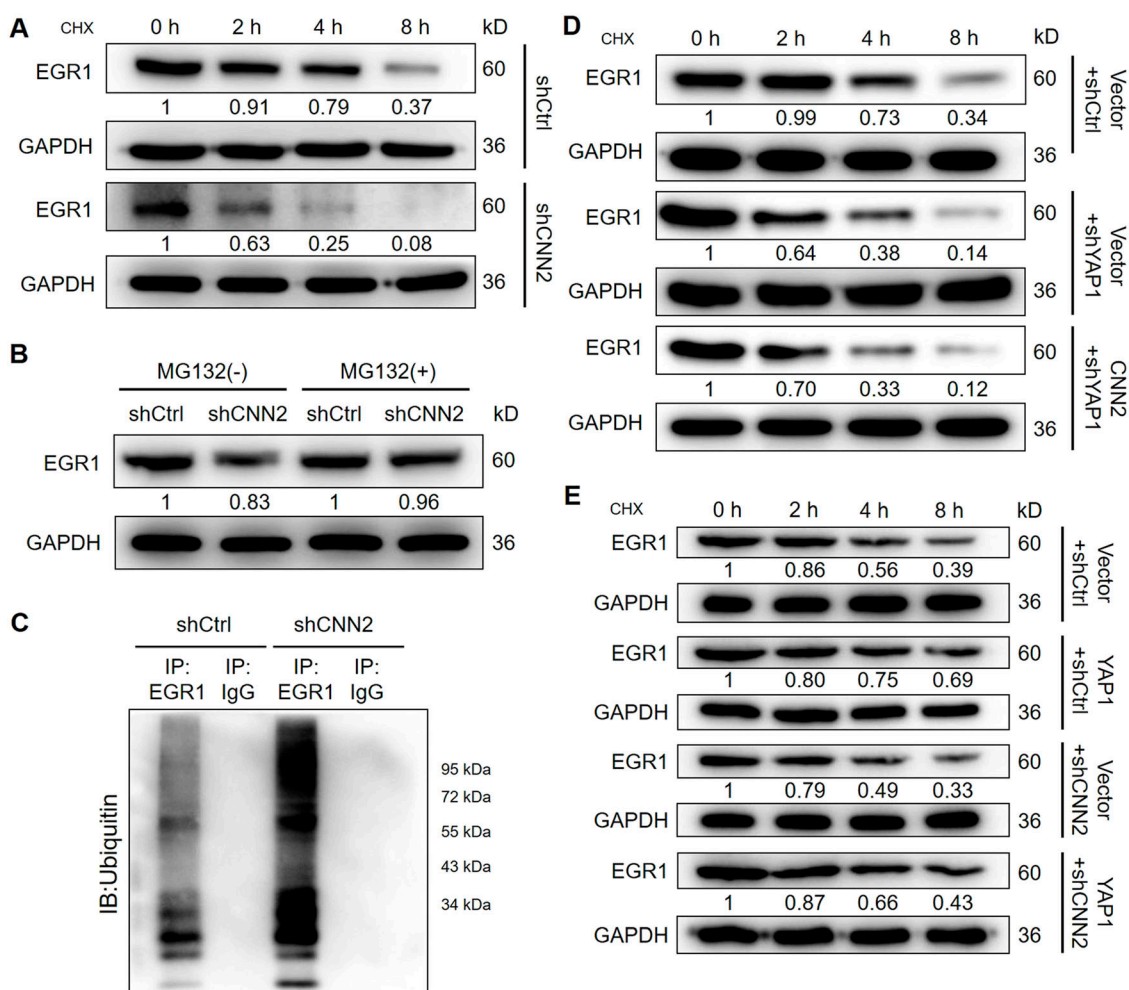

**Figure 5. CNN2 may regulate ubiquitination and the expression of EGR1 in a YAP1-dependent manner.**
**(A)** After the treatment of CHX (10 μM), shCtrl and shCNN2 RKO cells were lysed at indicated time points and subjected to Western blot to detect the expression of EGR1, evaluating the protein stability of EGR1. **(B)** Protein expression of EGR1 was detected by Western blot in shCtrl and shCNN2 cells with or without MG132 (10 μM) treatment. **(C)** Immunoprecipitated complex from shCtrl or shCNN2 using anti-EGR1 antibody or IgG was subjected to ubiquitin detection to assess the ubiquitination level of EGR1. **(D, E)** Protein stability of EGR1 was detected by the above-mentioned strategy in indicated cell models. The quantitative analysis was performed based on protein/GAPDH using ImageJ.
Source data are available for this figure.

its potential in the diagnosis and treatment of breast cancer (Ji et al, 2015). In gastric cancer, a typical loss-of-function study made on CNN2 revealed the suppression of tumor development upon CNN2 silence (Hu et al, 2017). Similar regulation of cancer cell phenotypes by CNN2 was also displayed in prostate cancer and hepatocellular carcinoma by Jin et al and Zhou et al, respectively (Moazzem Hossain et al, 2014; Kang et al, 2018). In this study, our in vitro and in vivo results also recognize CNN2 as a tumor promoter in the development and progression of CRC. The diminishment of CNN2 expression in CRC cells could significantly block cell proliferation in vitro and tumor growth in vivo. Otherwise, CNN2 also showed a regulatory effect on cell migration, combined with the correlation between CNN2 expression and lymph node metastasis in clinicopathological statistical analysis, which indicated that it also had a potential regulatory effect on tumor metastasis and is consistent with the previous work

concentrating the regulatory effects of CNN2 on CRC cell migration (Zheng et al, 2020).

EGR1 (early growth response 1) is a member of the early growth response gene family, whose expression is regulated by many extracellular signaling molecules. At the same time, EGR1 has a highly conserved DNA-binding domain, which consists of three Cys2His2 zinc finger structures and is able to specifically recognize and bind the 5′-GCGGGGGCG-3′ sequence in the promoter region of the target genes, thus exerting a role in the transcriptional regulation of them (Eto et al, 2006; Ao et al, 2019; Havis & Duprez, 2020). In recent years, a large number of studies have shown that the biological function of EGR1 is closely related to cell proliferation, apoptosis, migration, invasion, differentiation, and so on. In terms of tumor regulation, EGR1 promotes tumor progression by improving the growth and reproduction ability, as well as migration

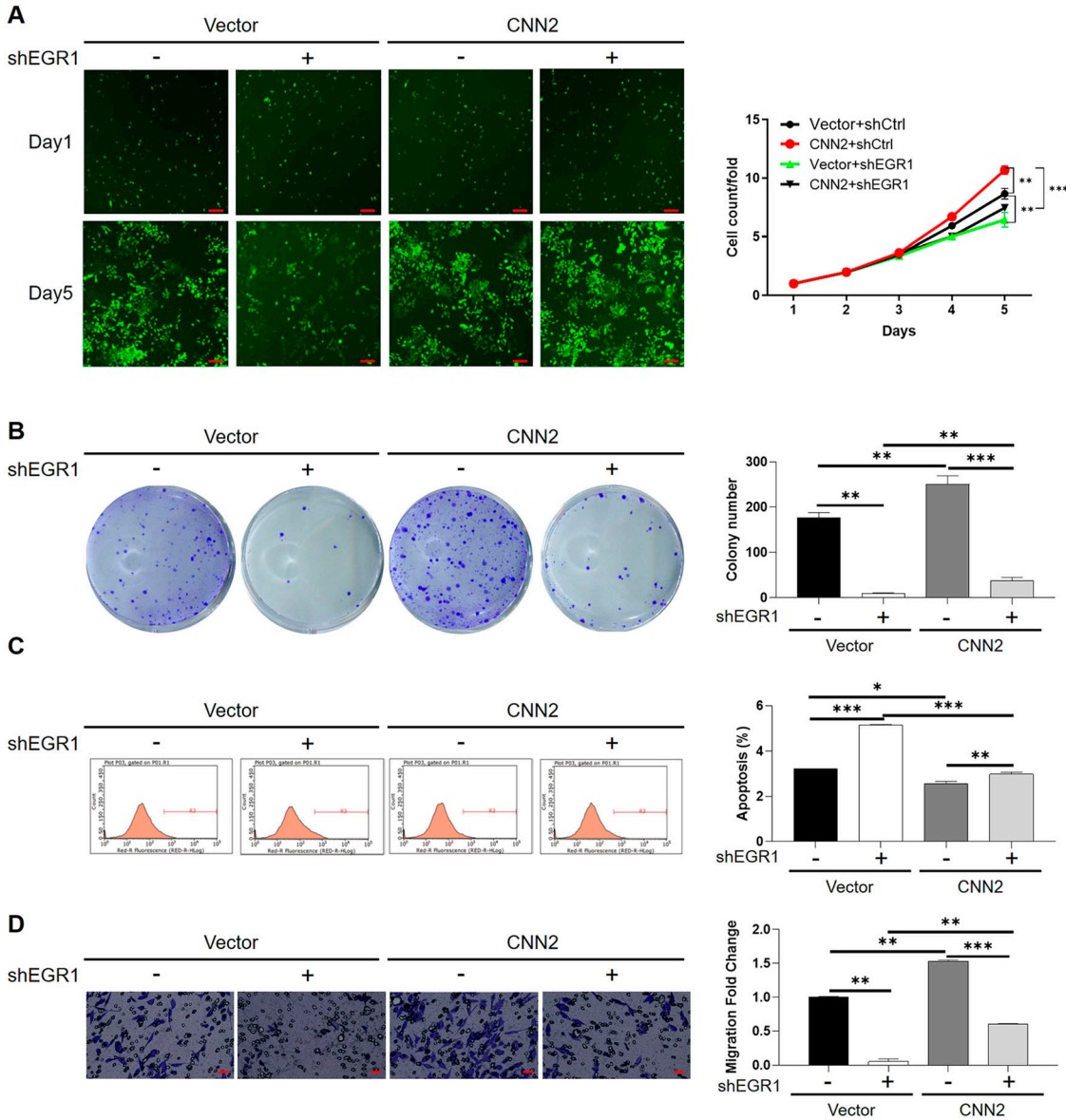

**Figure 6. EGR1 is essential for the CNN2-induced regulation of CRC.**
Lentivirus targeting CNN2 overexpression (CNN2) or EGR1 silencing (shEGR1) and the corresponding negative control (Vector or shCtrl) were used for RKO cell transfection. **(A)** Cell proliferation of RKO cells in different groups was detected by the Celigo cell counting assay. Scale bar = 50 $\mu$m. **(B)** Colony formation ability of RKO cells in different groups was detected by the colony formation assay. **(C)** Cell apoptosis of RKO cells in different groups was detected by flow cytometry. **(D)** Cell migration of RKO cells in different groups was detected by the Transwell assay. Scale bar = 50 $\mu$m. Data were shown as the mean ± SD. *$P < 0.05$, **$P < 0.01$, and ***$P < 0.001$. Source data are available for this figure.

and invasion ability, of cancer cells in some tumors, but plays a quite opposite role as tumor suppressor in others (Crawford et al, 2019; Li et al, 2019; Tong-Tong et al, 2019; Knudsen et al, 2020; Wang et al, 2021). Several pieces of evidence declared EGR1 as an oncogene-like molecule in gastric cancer. Huang et al illustrated that EGR1 may decrease the transcription of miR-195 through interacting DNMT3L, thus assisting gastric cancer cells to resist apoptosis (Yang et al, 2019). Ji et al clarified a similar role of EGR1 in gastric cancer through a different pathway, indicating that EGR1-induced activation of linc01503 caused the change in gastric cancer

cell phenotypes (Ma et al, 2021). Conversely, some forceful outcomes define EGR1 as a tumor suppressor in breast cancer. Wong et al suggested that EGR1, together with CTCF, exerted an inhibitory effect on cell migration and tumor metastasis, possibly through transcriptionally regulating Nm23-H1 (Wong et al, 2021). When the EGR1 signal was dampened in breast cancer by EZH2, cell growth, migration, and invasion, and tumorigenesis of breast cancer cells could be recovered or strengthened (Guan et al, 2020). In this study, EGR1 was screened as a potential downstream of CNN2, which possessed a similar expression pattern in CRC (higher in tumor

tissues than in normal tissues). Functional "rescue" experiments demonstrated that the apparent promotion effects of CRC development by CNN2 overexpression depend on the existence of EGR1 and are partially lost upon EGR1 silencing. Notably, it is surprising that we obtained paradoxical results with the previous study concerning the relationship between EGR1 and CRC, which proposed that RNF2 promotes CRC development through down-regulating EGR1 (Wei et al, 2020). Therefore, the amphibious functions of EGR1 in CRC are worth further investigation.

On the contrary, it was previously exhibited that EGR1 could form a complex with a Yes kinase-associated protein 1 (YAP1), thus regulating lipopolysaccharide-induced tissue factor expression in human endothelial cells or cell phenotypes of prostate cancer (Zagurovskaya et al, 2009; Yi et al, 2016). Herein, we not only proved the interaction between EGR1 and YAP1 in CRC cells, but also found the interaction between CNN2 and EGR1 or YAP1. On this basis, we supposed that CNN2 may influence the expression of EGR1 through forming the CNN2/YAP1/EGR1 complex. Moreover, it was previously reported that EGR1 expression could be affected by post-translational modifications such as SUMOylation and ubiquitination (Manente et al, 2011). Correspondingly, the outcomes of protein stability and protein ubiquitination examinations in this study suggested that CNN2 knockdown may down-regulate EGR1 expression through enhancing ubiquitination and proteasome-related protein degradation. More importantly, we also found that the CNN2-induced regulation of EGR1 protein stability exists in a YAP1-dependent manner.

In conclusion, CNN2 was discovered as a tumor promoter in CRC, which was up-regulated in CRC and could promote CRC development through regulating cell phenotypes. Mechanistically, it was found that CNN2 may form a complex with YAP1 and EGR1, thus regulating the expression of EGR1 and CRC. Therefore, this study identified a novel potential therapeutic target for CRC.

# Materials and Methods

### Tissue microarray and immunohistochemistry (IHC)

A tissue microarray containing CRC tissues and the normal tissues adjacent to the cancer was purchased from Shanghai Outdo Biotech Co., Ltd (No. HColA180Su17). Informed forms were signed by patients who provided tissues, and corresponding clinical information was provided as well. The tissue microarray chip was applied for IHC analysis with CNN2 and ERG1 antibodies. IHC representative images of CNN2 and ERG1 were captured, and the IHC scores were determined by the sum of staining intensity scores and staining extent scores. The antibodies used are listed in Table S2. This research was approved by the Ethics Committee of Changhai Hospital Affiliated to Navy Medical University.

### Cell culture

Human colon carcinoma cell lines applied in our study including RKO and HCT116 were purchased from Cell Resource Center, Institute of Basic Medicine, Chinese Academy of Medical Sciences. All

cells were cultured in 1640 medium containing 10% FBS in a cell incubator with 5% $CO_2$ at 37°C.

### Infection of target cells by lentivirus

The overexpression vector of CNN2 or EGR1 and short hairpin RNA for CNN2, YAP1, and ERG1 were constructed in Shanghai YBR Bioscires. Co., Ltd, and the sequences are available in Table S3. For constructing cell models, lentivirus with target genes (2–5×10$^8$ TU/ml) was seeded into cells and cultured for 72 h. The fluorescence of cells was observed by a microscope, and target gene expression was confirmed by qRT-PCR and Western blot analysis. Stable gene expression cells were used for the following experiments.

### qRT–PCR

Total RNA was extracted with the TRIzol reagent (Sigma-Aldrich) from cells, and cDNA was obtained by reverse transcription using Promega M-MLV Kit (Promega Corporation). SYBR Green Master Mix Kit (Vazyme) was used for real-time qRT-PCR with GAPDH as a housekeeping gene. The relative expression level of RNA was calculated by the $2^{-\triangle\triangle Ct}$ method. The primer sequences applied in qRT-PCR are detailed in Table S4.

### Western blotting and Co-IP

Cells were lysed for total protein extraction; next, total proteins were quantified using the BCA method. Total protein (20 μg) segregation was accomplished by 12% SDS–PAGE and PVDF membrane. The membrane was blocked with TBST solution containing 5% skimmed milk and then incubated with primary antibodies overnight at 4°C. After being washed with the TBST solution three times for 10 min each, the corresponding secondary antibody was added. The protein signals were evaluated by an enhanced chemiluminescence detection system. For Co-IP analysis, total proteins were immunoprecipitated by CNN2 and EGR1 or YAP1 antibody or immunoprecipitated by EGR1 and YAP1, and analyzed by Western blotting. The antibodies used are available in Table S2. The quantitative analysis was performed based on protein/GAPDH using ImageJ.

### CCK8 assay

Cell suspension (100 μl/well) was added into a 96-well plate, which was cultured at 37°C with 5% $CO_2$. Before performing the detection, 10 μl of Cell Counting Kit-8 solution was added to each well and the cells were cultured at 37°C for a further 2 h. The absorbance at the wavelength of 450 nm was measured with a microplate reader for evaluating the cell viability.

### Celigo cell counting assay

The cell proliferation potential was detected using the Celigo cell counting assay. Cells (2.5 × 10$^3$ per well) were seeded into 96-well plates and cultured in 1640 medium plus 10% FBS for 5 d. Meanwhile, the culture medium was replaced with a fresh medium every

3 d. Cell counts were measured by a Celigo image cytometer on days 1, 2, 3, 4, and 5 at the same time, and the data were analyzed.

### FACS

Apoptosis analysis was carried out by dye staining using Annexin V-APC/PI. When the fusion degree reached 70%, the cells were harvested, centrifuged at $300g$ for 5 min, washed with 4°C pre-cooled D-Hanks, and stained with Annexin V-APC. After the staining of PI, the stained cells were subjected to a FACSCanto II Flow Cytometry to evaluate the cell apoptosis level.

### Wound-healing assay

Lentivirus-infected HCT116 and RKO cells ($5 \times 10^4$ cells/well) were seeded into 96-well plates. When the confluence of the cells reached 90%, the medium was substituted with a low-concentration serum medium and scratched on the cell layer using 96 wounding replicators (VP Scientific). Then, the cells were washed two to three times with serum-free medium and incubated in an incubator with 5% $CO_2$ at 37°C. Photographs of the wound were captured by a fluorescence microscope at 0, 24, and 48 h. Finally, the migration area was analyzed with Cellomics.

### Transwell assay

The Transwell chamber assay (3422; Corning) was used for the migration assay. The transfected HCT116 and RKO cells were resuspended in a low-serum medium and were seeded on the upper chambers without serum. Besides, to the lower chamber was added 600 $\mu l$ culture medium containing 30% FBS (with serum). After 24 h, the non-metastatic cells in the upper chamber were gently removed by a cotton swab with PBS. Then, the upper chamber was immersed in the staining solution with crystal violet for 5 min. After washing and drying, the chamber was taken photographs under the microscope.

### Colony formation assay

800 lentivirus-infected cells were seeded into a six-well plate (2 ml/well) and cultured for 8 d with medium changed every 3 d. Visible clones were recorded by a fluorescence microscope. After PBS washing, clones were fixed with 4% paraformaldehyde and stained by Giemsa (Dingguo). Pictures were collected, and the clones' number was recorded.

### Xenograft tumor model

24-wk-old nude mice were randomly divided into two groups with 10 in each, which were purchased from Beijing Vital River Laboratory Animal Technology Co., Ltd. shCNN2 RKO cells or control cells were subcutaneously injected into nude mice ($4 \times 10^6$ cells for each). After 5 wk, each animal was weighed, and the length and diameter of tumor were measured, and the data were continuously collected for 4 wk. At the last time of measuring, 0.7% sodium pentobarbital (10 $\mu l/g$) was intraperitoneally injected into all mice. The in vivo imaging system (IVIS Spectrum; PerkinElmer) was used to observe the fluorescence intensity. Then, mice were euthanized and tumor tissues were collected. Then, tumor tissues were paraffin-embedded to cut into 4-$\mu m$ slides for Ki-67 staining analysis with a Ki-67 antibody. This research was approved by the Ethics Committee of Changhai Hospital Affiliated to Navy Medical University.

### Human gene expression array

Gene expression in shCNN2 RKO cells and control RKO cells was detected using Human Gene Expression Array. Firstly, total RNA was extracted and RNA quality and integrity were determined by NanoDrop 2000 (Thermo Fisher Scientific) and Agilent 2100 and Agilent RNA 6000 Nano Kit (Agilent). Then, RNA sequencing was performed with Affymetrix human GeneChip PrimeView according to the instructions. The outcomes were scanned by Affymetrix Scanner 3000. Raw data filtering was completed using a Welch $t$ test with the Benjamini–Hochberg FDR (|fold change| ≥ 1.3 and FDR < 0.05 as significant). Finally, the Ingenuity Pathway Analysis (QIAGEN) was executed and |Z-score| ≥ 2 is considered valuable.

### Statistical analysis

All experiments were in triplicate, and data were analyzed using GraphPad Prism 8 and SPSS 19.0 (IBM SPSS) and presented as the mean ± SD. The differences were analyzed using a $t$ test between the two groups; if not, one-way ANOVA followed by Bonferroni's post hoc test was used. IHC data were expressed as the median, and a paired $t$ test was used to examine the statistical significance. The Mann–Whitney U analysis, the Spearman rank correlation coefficient, and the Kaplan–Meier survival analysis were used to assess the relationship between the expression of CNN2 and characteristics of CRC patients. $P < 0.05$ was considered to be significantly different.

## Data Availability

The data that support the findings of this study are available in the NCBI Gene Expression Omnibus (GEO) database, with the accession number GSE229716.

### Ethics Statement

This research was approved by the Ethics Committee of Changhai Hospital Affiliated to Navy Medical University.

## Supplementary Information

## Acknowledgements

None.

### Author Contributions

J He: conceptualization, validation, investigation, methodology, and writing—original draft.

X Yang: conceptualization, validation, investigation, methodology, and writing—original draft.

C Zhang: conceptualization, software, investigation, and methodology.

A Li: formal analysis, investigation, and methodology.

W Wang: resources, formal analysis, and visualization.

J Xing: resources, data curation, and software.

J E: formal analysis and methodology.

X Xu: resources and software.

H Wang: formal analysis and methodology.

E Yu: resources and formal analysis.

D Shi: conceptualization, data curation, supervision, and writing—review and editing.

H Wang: validation, visualization, project administration, and writing—review and editing.

### Conflict of Interest Statement

The authors declare that they have no conflict of interest.

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
