## [Reviewer comments · Life Science Alliance]

Life Science Alliance

CNN2 silencing inhibits colorectal cancer development through promoting ubiquitination of EGR1

Jinghu He, Xiaohong Yang, Chuansen Zhang, Ang Li, Wei Wang, Junjie Xing, Jifu E, Xiaodong Xu, Hao Wang, Enda Yu, Debing Shi and Hantao Wang

DOI: <https://doi.org/10.26508/lsa.202201639>

Corresponding author(s): Dr. Jinghu He (Changhai Hospital)

Review Timeline:

Submission Date:	2022-07-29
Editorial Decision:	2022-10-10
Revision Received:	2023-03-03
Editorial Decision:	2023-04-12
Revision Received:	2023-04-20
Accepted:	2023-04-21

Scientific Editor: Novella Guidi

Transaction Report:

October 10, 2022

Re: Life Science Alliance manuscript #LSA-2022-01639

Dr. Jinghu He
Changhai Hospital
Changhai Hospital Affiliated to Navy Medical University, Shanghai, China.
Shanghai 200433
China

Dear Dr. He,

Thank you for submitting your manuscript entitled "CNN2 silencing inhibits colorectal cancer development through regulating ubiquitination of EGR1" to Life Science Alliance. The manuscript was assessed by expert reviewers, whose comments are appended to this letter. We invite you to submit a revised manuscript addressing the Reviewer comments.

Thank you for this interesting contribution to Life Science Alliance. We are looking forward to receiving your revised manuscript.

Sincerely,

B. MANUSCRIPT ORGANIZATION AND FORMATTING:

Reviewer #1 (Comments to the Authors (Required)):

In this manuscript, He et al examined the expression of CNN2 in CRC, and studied the mechanism how CNN2 promotes CRC progression. They showed that CNN2 was highly expressed in CRC, and CNN2 could promote CRC progression through inhibiting the ubiquitination and degradation of EGR1 in a YAP1 dependent manner. The manuscript was written clearly, and results were logically displayed. However, there still have some problems that need to be clarified or revised, shown as following.

1. In the title, the word "regulating ubiquitination" need to be changed to "promoting (or enhancing) ubiquitination and degradation", to make it clear.
2. In the result part of "CNN2 is upregulated in colorectal cancer and associated with disease development", they say "similar upregulation of CNN2", here the result from TCGA database is mRNA level, not protein level. Should describe more clearly.
3. The full name of "colorectal cancer" needs to be changed to CRC, only need to provide the first time it appeared.
4. In the result part of "Silencing CNN2 inhibits colorectal cancer development in vivo", the description "xenografts in shCtrl group truly showed more vigorous growth activity" should describe the shCNN2 group, not the control group.
5. In the result part of "CNN2 may regulate colorectal cancer development through affecting EGR1 expression", they say "Among the several candidates with downregulated expression in outcomes of both qPCR and western blotting, EGR1 was found to be upregulated", the description may bring confusion, should clarify "the downregulated expression in CNN2 knockdown cells".
6. Figure 6D is not wound-healing, but transwell? Figure legend needs to be revised.
7. They used three shRNAs to knockdown CNN2, but only provide the phenotype of only one shCNN2, results of at least two shRNAs should to be provided.
8. In Figure 4C, the control means "IgG"? This needs to be clarified. If control means IgG, why IgG IP group had CNN2 and EGR1 bands in IB, but not YAP1?
9. They used many same figures 4C and 4E, if these data came from one experiment, they need to be displayed in the same panel, no need to separate them into two figures.
10. The labeling in Figure 4D is confusing, I can't understand that.
11. In figure 5A and 5D, they need to do quantitative analysis, to make the data more clearly.
12. The design in 5D is confusing, they should overexpress YAP1 in CNN2 knockdown group, even they overexpress CNN2, they should provide the data of CNN2 overexpression only group. In addition, the function of YAP1 in the degradation of EGR1 is not clear from current data.
13. The labeling in Figure 5B can use a line to separate the MG132(-) and MG132(+) group.
14. They need to draw a schematic figure to show the proposed mechanism, so the readers could understand the mechanism more easily.
15. The rescue experiments in Figure 6 were not logical, they should overexpress EGR1 in shCNN2 group to confirm that CNN2 indeed promote CRC progression through degradating EGR1. Even they used overexpressing CNN2 in CRC cells, I didn't see the expression of CNN2 and EGR1 in protein level.
16. According to RT-PCR data in Figure S7, it seems that the expression of EGR1 was upregulated in CNN2-overexpressing group, does this mean that CNN2 could regulate EGR1 expression in mRNA level. Similarly, it seems that shEGR1 could also inhibit the expression of CNN2, does this means EGR1 could also regulate the expression of CNN2?
17. The expression and description in some sentences are not conformed with English style, need to be revised. Would like to suggest the authors to polish the language.

Reviewer #2 (Comments to the Authors (Required)):

1. Since CNN protein family has at least 3 members, it was not clearly justified why CNN2 was chosen for the study. Do other members of CNN have the same or similar effects on CRC cells?
2. The role of CNN2 has been studied (Ectopic expression of CNN2 of colon cancer promotes cell migration), why not mentioned and compared in this paper?
3. In table 2, how is the relationship between CNN2 expression and distant metastasis.
4. In table 2, what do stage and grade need to be specified?

5. In Figure 1A, why no Tumor (Stage II).
6. It is recommended to add a histogram: the relationship between CNN2 expression and Normal, Tumor (Stage I), Tumor (Stage II), Tumor (Stage III), and Tumor (Stage IV)
7. Kaplan-Meier analysis was based on IHC results and TCGA results?
8. In figure S1, there is only one group of experimental result for knockdown efficiency. Which cells were tested for knockdown efficiency? how about another?
9. In figure S2, why is the transfection efficiency of the shCNN2 group so low but has highly interference effect?
10. figure S2 can be merged with figure S1.
11. CCF1 and GPX4 are also downregulated. Why not select for these two genes?
12. Gene Expression Profiling should be uploaded to a public database, such as GEO.
13. The expression of CNN2 is different in Figure S5 and Figure 2A. Please conduct statistical analysis on the expression of CNN2 protein.
14. In Figure S7, why did shEGR1 transfection reduce the expression of CNN2?
15. To prove that EGR1 is downstream of CNN2, rescue experiments should be performed, that is shCNN2+EGR1 overexpression.

Response to Editor:

1. As required, we addressed all the comments and questions from the reviewers item by item. In our Response Letter, we inserted our responses after the reviewer's comments, marked by the phrase "response".
2. As required, we upload a revised manuscript as the 'revised Manuscript' file. All the change could be found through the change tracking system of Microsoft Word.
3. Otherwise, we have uploaded the revised figures as required.
4. Notably, considering the key guidance provided by Debing Shi (Department of Colorectal Surgery, Fudan University Shanghai Cancer Center, 270 Dong'an Road, Shanghai, China; Department of Oncology, Shanghai Medical College, Fudan University, Shanghai, China), we decided to enlist him as a corresponding author in our manuscript. Revision has been made to the author list of the manuscript.

Reviewer #1 (Comments to the Authors (Required)):

In this manuscript, He et al examined the expression of CNN2 in CRC, and studied the mechanism how CNN2 promotes CRC progression. They showed that CNN2 was highly expressed in CRC, and CNN2 could promote CRC progression through inhibiting the ubiquitination and degradation of EGR1 in a YAP1 dependent manner. The manuscript was written clearly, and results were logically displayed. However, there still have some problems that need to be clarified or revised, shown as following.

1. In the title, the word "regulating ubiquitination" need to be changed to "promoting (or enhancing) ubiquitination and degradation", to make it clear.

Response: Thanks for the kind suggestion from Reviewer.

As suggested, we have revised the title as "CNN2 silencing inhibits colorectal cancer development through promoting ubiquitination of EGR1".

2. In the result part of "CNN2 is upregulated in colorectal cancer and associated with disease development", they say "similar upregulation of CNN2", here the result from TCGA database is mRNA level, not protein level. Should describe more clearly.

Response: Thanks for the comments from the reviewer.

As suggested, we have made corresponding revision to make the description more clearly.

3. The full name of "colorectal cancer" needs to be changed to CRC, only need to provide the first time it appeared.

Response: Greatly thank the kind suggestion from the reviewer.

As suggested, we have changed most of "colorectal cancer" to CRC.

4. In the result part of "Silencing CNN2 inhibits colorectal cancer development in vivo", the description "xenografts in shCtrl group truly showed more vigorous growth activity" should describe the shCNN2 group, not the control group.

Response: Thank the reviewer for his suggestion.

Herein, after checking, we confirmed that the description is right. The xenografts in shCtrl group actually showed more activity in growth and grew faster than that in shCNN2 group.

5. In the result part of "CNN2 may regulate colorectal cancer development through affecting EGR1 expression", they say "Among the several candidates with downregulated expression in outcomes of both qPCR and western blotting, EGR1 was found to be upregulated", the description may bring confusion, should clarify "the downregulated expression in CNN2 knockdown cells".

Response: Thanks the comment from the reviewer.

As suggested, revision has been made for more precise description.

6. Figure 6D is not wound-healing, but transwell? Figure legend needs to be revised.

Response: Thanks for the comments from the reviewer.

We are sorry for the mistake, which has been corrected in the revised version.

7. They used three shRNAs to knockdown CNN2, but only provide the phenotype of only one shCNN2, results of at least two shRNAs should to be provided.

Response: Greatly thank the kind suggestion from the reviewer.

As suggested, we have repeated the loss-of-function assays of CNN2 with 2 shRNAs (revised Figure 2).

8. In Figure 4C, the control means "IgG"? This needs to be clarified. If control means IgG, why IgG IP group had CNN2 and EGR1 bands in IB, but not YAP1?

Response: Thanks the comment from the reviewer.

We apologize for the misunderstanding brought by the wrong labelling. Herein, we have revised the "control" to "vector", which represent the cells transfected with negative control lentivirus.

9. They used many same figures 4C and 4E, if these data came from one experiment, they need to be displayed in the same panel, no need to separate them into two figures.

Response: Greatly appreciate the careful examination from the reviewer.

As suggested, we have combined Figure 4C and 4E into Figure 4C.

10. The labeling in Figure 4D is confusing, I can't understand that.

Response: Thanks for the reviewer's comments.

We are sorry for the inconvenience. Herein, the labeling has been revised for easier understanding.

11. In figure 5A and 5D, they need to do quantitative analysis, to make the data more clearly.

Response: Thank the reviewer for his suggestion.

As suggested, we have added the quantitative analysis for figure 5.

12. The design in 5D is confusing, they should overexpress YAP1 in CNN2 knockdown group, even they overexpress CNN2, they should provide the data of CNN2 overexpression only group. In addition, the function of YAP1 in the

degradation of EGR1 is not clear from current data.

Response: Greatly thank the kind suggestion from the reviewer.

As suggested, we added another rescue assay of CNN2 knockdown and YAP1 overexpression for a more clear demonstration of the CNN2-induced and YAP1-dependent regulation of EGR1 protein stability.

13. The labeling in Figure 5B can use a line to separate the MG132(-) and MG132(+) group.

Response: Thanks for the reviewer's comments.

As suggested, corresponding revision has been made.

14. They need to draw a schematic figure to show the proposed mechanism, so the readers could understand the mechanism more easily.

Response: Thanks for the reviewer's comments.

As suggested, a schematic figure has been added.

15. The rescue experiments in Figure 6 were not logical, they should overexpress EGR1 in shCNN2 group to confirm that CNN2 indeed promote CRC progression through degrading EGR1. Even they used overexpressing CNN2 in CRC cells, I didn't see the expression of CNN2 and EGR1 in protein level.

Response: Thanks the kind suggestion from the reviewer.

As suggested, we have supplemented a rescue assay of CNN2 knockdown and EGR1 overexpression for better demonstrating the synergistic regulation of CRC by CNN2/EGR1 axis. Moreover, the detection of CNN2 and EGR1 on mRNA and protein levels in different groups of cells was also repeated or supplemented.

16. According to RT-PCR data in Figure S7, it seems that the expression of EGR1 was upregulated in CNN2-overexpressing group, does this mean that CNN2 could regulate EGR1 expression in mRNA level. Similarly, it seems that shEGR1 could also inhibit the expression of CNN2, does this means EGR1 could also regulate the expression of CNN2?

Response: Greatly appreciate the positive comments from the reviewer.

Actually, in some experiments, it seems that EGR1 also has some regulatory effects on CNN2. However, in this study, we want to concentrate our study on the effects of

CNN2/EGR1 axis on CRC development. Therefore, we have revised the modes of presentation of the CNN2 and EGR1 expression in different groups of cells in supplementary materials. On the other hand, in our future work, we would like to explore the possibility of CNN2 and EGR1 to form feedback loop in the regulation of CRC.

17. The expression and description in some sentences are not conformed with English style, need to be revised. Would like to suggest the authors to polish the language.

Response: Thanks the comment from the reviewer.

On the occasion, we have carefully checked the manuscript again, and polished the Manuscript English accordingly.

Reviewer #2 (Comments to the Authors (Required)):

1. Since CNN protein family has at least 3 members, it was not clearly justified why CNN2 was chosen for the study. Do other members of CNN have the same or similar effects on CRC cells?

Response: Thanks for the reviewer's comments.

As suggested, we supplemented bioinformatics analysis on the differential expression of CNN1 and CNN3 in colorectal cancer tissues and normal tissues based on TCGA database. It is clear that CNN2 was also the only one with upregulation in CRC tissues among the 3 members of CNN family

2. The role of CNN2 has been studied (Ectopic expression of CNN2 of colon cancer promotes cell migration), why not mentioned and compared in this paper?

Response: Great appreciation for the kind suggestion.

As suggested, we have included this reference in the discussion section. Actually, the results in our study were in perfectly agreement with this study.

3. In table 2, how is the relationship between CNN2 expression and distant metastasis.

Response: Thanks the comment from the reviewer.

As indicated in the Materials and Methods section, the tissue microarray was purchased from commercial source. According to the appendant information of the

tissue microarray, almost all patients have no distant metastasis. Therefore, the statistical analysis could not be done. In our future work, we would collect more samples for demonstrating the correlation between CNN2 expression and distant metastasis of CRC.

4. In table 2, what do stage and grade need to be specified?

As indicated in the Materials and Methods section, the tissue microarray was purchased from commercial source. According to the appendant information of the tissue microarray, stage and grade mean AJCC clinical stage and pathological grade, respectively.

5. In Figure 1A, why no Tumor (Stage II).

Response: Thanks for the reviewer's comments.

As suggested, typical images of Tumor (Stage II) have been supplemented.

6. It is recommended to add a histogram: the relationship between CNN2 expression and Normal, Tumor (Stage I), Tumor (Stage II), Tumor (Stage III), and Tumor (Stage IV)

Response: Great appreciation for the kind suggestion.

As suggested, we attempt to make a histogram for displaying the relationship between CNN2 expression with different types of tissues. However, although the statistical analysis indicated that there is a significant correlation between CNN2 expression with tumor stage, the histogram was not so good to be presented in the manuscript.

7. Kaplan-Meier analysis was based on IHC results and TCGA results?

Response: Thanks for the reviewer's nice suggestion.

The survival analysis was performed based on IHC results obtained from the tissue microarray.

8. In figure S1, there is only one group of experimental result for knockdown efficiency. Which cells were tested for knockdown efficiency? how about another?

Response: Thanks the kind suggestion from the reviewer.

As suggested, we supplemented the detection of knockdown efficiency in both cell lines and obtained similar results.

9. In figure S2, why is the transfection efficiency of the shCNN2 group so low but has

highly interference effect?

Response: We greatly appreciated the constructive comments from the reviewer.

Actually, the transfection efficiency of the shCNN2 group was not so low. According to the living cell imaging, the transfection efficiencies of both shCtrl and shCNN2 lentiviruses are above 80%, which is considered to be a successful transfection.

10. figure S2 can be merged with figure S1.

Response: Thanks for the reviewer's nice suggestion.

Considering that these two figures showed totally different results, we prefer to display them separately for avoiding any possible misunderstanding.

11. CCNF and GPX4 are also downregulated. Why not select for these two genes?

Response: Thanks the comment from the reviewer.

As shown in the revised figure S5E, among the 3 candidates, only EGR1 showed a co-expression profile with CNN2. Corresponding description has also been added in the manuscript.

12. Gene Expression Profiling should be uploaded to a public database, such as GEO.

Response: Great thanks to reviewer's constructive suggestion.

Actually, we planned to upload our gene expression profiling onto GEO database after the acceptance of our manuscript.

13. The expression of CNN2 is different in Figure S5 and Figure 2A. Please conduct statistical analysis on the expression of CNN2 protein.

Response: Greatly appreciate the careful examination from the reviewer.

Actually, the intensity of WB plot could be affected by a lot of factors including the time of exposure. Therefore, the intensities of WB plot in two separate experiments could not be compared. For both figures, it could be clearly illustrated that the interference of CNN2 expression was successful.

14. In Figure S7, why did shEGR1 transfection reduce the expression of CNN2?

Response: Greatly appreciate the positive comments from the reviewer.

Actually, in some experiments, it seems that EGR1 also has some regulatory effects on CNN2. However, in this study, we want to concentrate our study on the effects of CNN2/EGR1 axis on CRC development. Therefore, we have revised the modes of

presentation of the CNN2 and EGR1 expression in different groups of cells in supplementary materials. On the other hand, in our future work, we would like to explore the possibility of CNN2 and EGR1 to form feedback loop in the regulation of CRC.

15. To prove that EGR1 is downstream of CNN2, rescue experiments should be performed, that is shCNN2+EGR1 overexpression.

Response: Thanks the kind suggestion from the reviewer.

As suggested, we have supplemented a rescue assay of CNN2 knockdown and EGR1 overexpression for better demonstrating the synergistic regulation of CRC by CNN2/EGR1 axis. Moreover, the detection of CNN2 and EGR1 on mRNA and protein levels in different groups of cells was also repeated or supplemented.

April 12, 2023

RE: Life Science Alliance Manuscript #LSA-2022-01639R

Dr. Jinghu He
Changhai Hospital
Changhai Hospital Affiliated to Navy Medical University, Shanghai, China.
Shanghai 200433
China

Dear Dr. He,

Thank you for submitting your revised manuscript entitled "CNN2 silencing inhibits colorectal cancer development through promoting ubiquitination of EGR1". We would be happy to publish your paper in Life Science Alliance pending final revisions necessary to meet our formatting guidelines.

- please deposit your Gene Expression Profiling to a public database, such as GEO and add a data availability section in the manuscript in which you'll provide the accession number
- please add ORCID ID for corresponding author-you should have received instructions on how to do so
- please add Keywords for your manuscript to our system
- please add the Twitter handle of your host institute/organization as well as your own or/and one of the authors in our system
- please define what your 'Related Manuscript Files' are. If they are source data for any figure, please combine them according to the related Figure

Figure Check:

- please add scale bars to Figure 2E, Figure 3C, Figure 6 A and D, Figure S3, Figure S13
- please add molecular weights next to all blots (both in main Figures and Supp. Figs.)
- please rename Figure S14 as a graphical abstract

A. FINAL FILES:

B. MANUSCRIPT ORGANIZATION AND FORMATTING:

Sincerely,

-please deposit your Gene Expression Profiling to a public database, such as GEO and add a data availability section in the manuscript in which you'll provide the accession number

Response: Thanks for your comments, the GEO accession number is GSE229716.

-please add ORCID ID for corresponding author-you should have received instructions on how to do so

Response: ORCID of Debing Shi is 0000-0001-7980-8886

-please add Keywords for your manuscript to our system

Response: Thanks for your comments, them have been added on the submission

system.

-please add the Twitter handle of your host institute/organization as well as your own or/and one of the authors in our system

Response: We are sorry for that we cannot register Twitter account.

-please define what your 'Related Manuscript Files' are. If they are source data for any figure, please combine them according to the related Figure

Response: Thanks for your comments, it has been revised accordingly.

Figure Check:

-please add scale bars to Figure 2E, Figure 3C, Figure 6 A and D, Figure S3, Figure S13

Response: Thanks for your comments, these figures have been revised accordingly.

-please add molecular weights next to all blots (both in main Figures and Supp. Figs.)

Response: Thanks for your comments, these figures have been revised accordingly.

-please rename Figure S14 as a graphical abstract

Response: Thanks for your comments, it has been revised accordingly.

April 21, 2023

RE: Life Science Alliance Manuscript #LSA-2022-01639RR

Dr. Jinghu He
Changhai Hospital
Changhai Hospital Affiliated to Navy Medical University, Shanghai, China.
Shanghai 200433
China

Dear Dr. He,

Thank you for submitting your Research Article entitled "CNN2 silencing inhibits colorectal cancer development through promoting ubiquitination of EGR1". It is a pleasure to let you know that your manuscript is now accepted for publication in Life Science Alliance. Congratulations on this interesting work.

DISTRIBUTION OF MATERIALS:

Again, congratulations on a very nice paper. I hope you found the review process to be constructive and are pleased with how the manuscript was handled editorially. We look forward to future exciting submissions from your lab.

Sincerely,
